# Prognostic Impact of *TERT* Promoter Mutations in Adult-Type Diffuse Gliomas Based on WHO2021 Criteria

**DOI:** 10.3390/cancers16112032

**Published:** 2024-05-27

**Authors:** Yujin Lee, Chul-Kee Park, Sung-Hye Park

**Affiliations:** 1Department of Hospital Pathology, St. Vincent’s Hospital, The Catholic University of Korea College of Medicine, 93, Jungbu-daero, Paldal-gu, Suwon 16247, Gyeonggi-do, Republic of Korea; yujinlee@catholic.ac.kr; 2Department of Neurosurgery, Seoul National University College of Medicine, 103 Deahak-ro, Jongno-gu, Seoul 03080, Republic of Korea; 3Department of Pathology, Seoul National University College of Medicine, 103 Deahak-ro, Jongno-gu, Seoul 03080, Republic of Korea; 4Neuroscience Institute, Seoul National University College of Medicine, 103 Deahak-ro, Jongno-gu, Seoul 03080, Republic of Korea

**Keywords:** genetics, glioblastoma, IDH-mutant astrocytoma, oligodendroglioma, *MGMT* promoter, *TERT* promoter

## Abstract

**Simple Summary:**

Telomerase reverse transcriptase promoter (TERTp) mutation is commonly observed in brain tumors and are known to contribute to the acquisition of immortality in tumors via maintaining telomere length. In this study, we aimed to investigate the prognostic impact of several known prognostic factors, including TERTp mutations, in 528 adult-type diffuse gliomas classified according to the 2021 WHO criteria. Our data showed that TERTp mutation status had a significant impact on prognosis in the combined group of glioblastoma, IDH-wildtype and astrocytoma, IDH-mutant, but was not a predictor of prognosis within any individual tumor groups. We also showed that several known clinicopathologic factors have different prognostic significance depending on the type of tumor. This supports the need for systematic tumor diagnosis based on molecular pathology classification and indicates that multiple factors, not just TERTp mutation status, should be considered in the prognosis of tumors.

**Abstract:**

Mutation in the telomerase reverse transcriptase promoter (*TERTp* )is commonly observed in various malignancies, such as central nervous system (CNS) tumors, malignant melanoma, bladder cancer, and thyroid carcinoma. These mutations are recognized as significant poor prognostic factors for these tumors. In this investigation, a total of 528 cases of adult-type diffuse gliomas diagnosed at a single institution were reclassified according to the 2021 WHO classifications of CNS tumors, 5th edition (WHO2021). The study analyzed clinicopathological and genetic features, including *TERTp* mutations in each tumor. The impact of known prognostic factors on patient outcomes was analyzed through Kaplan–Meier survival and Cox regression analysis. *TERTp* mutations were predominantly identified in 94.1% of oligodendrogliomas (ODG), followed by 66.3% in glioblastoma, IDH-wildtype (GBM-IDHwt), and 9.2% of astrocytomas, IDH-mutant (A-IDHm). When considering A-IDHm and GBM as astrocytic tumors (Group 1) and ODGs (Group 2), *TERTp* mutations emerged as a significant adverse prognostic factor (*p* = 0.013) in Group 1. However, within each GBM-IDHwt and A-IDHm, the presence of *TERTp* mutations did not significantly impact patient prognosis (*p* = 0.215 and 0.268, respectively). Due to the high frequency of *TERTp* mutations in Group 2 (ODG) and their consistent prolonged survival, a statistical analysis to evaluate their impact on overall survival was deemed impractical. When considering *MGMTp* status, the combined *TERTp*-mutated and *MGMTp*-unmethylated group exhibited the worst prognosis in OS (*p* = 0.018) and PFS (*p* = 0.034) of GBM. This study confirmed that the classification of tumors according to the WHO2021 criteria effectively reflected prognosis. Both uni- and multivariate analyses in GBM, age, *MGMTp* methylation, and *CDKN2A*/*B* homozygous deletion were statistically significant prognostic factors while in univariate analysis in A-IDHm, grade 4, the Ki-67 index and *MYCN* amplifications were statistically significant prognostic factors. This study suggests that it is important to classify and manage tumors based on their genetic characteristics in adult-type diffuse gliomas.

## 1. Introduction

The complex relationship between molecular genetic profile and the prognosis of gliomas underscores the importance of precise diagnostic criteria in the management and treatment of these tumors. The distinction between glioblastoma, IDH-wildtype (GBM-IDHwt), and oligodendroglioma (ODG) based on somatic mutations such as those in the telomerase reverse transcriptase promoter (TERTp) and isocitrate dehydrogenase (*IDH*) genes is particularly noteworthy.

Eukaryotic linear chromosomes have repetitive noncoding G-rich sequences called telomeres that protect natural DNA ends from DNA double-strand break repair machinery [1,2]. Telomeres shorten with each cell division and eventually lead to cellular senescence. In contrast, cancer cells have variable strategies to evade this cellular senescence and death by maintaining their telomere lengths [3]. One primary mechanism of telomere lengthening is the activation of telomerase, an enzyme that elongates the repeat sequences to the ends of chromosomes [4,5]. *TERT* is a catalytic protein subunit of telomerase and plays a crucial role in this process [5,6]. *TERTp* mutations lead to enhanced expression and activity of telomerase, thus contributing to cellular immortality, a hallmark of cancer [7,8,9]. Furthermore, TERT expression can be driven by various mechanisms such as amplification, chromosomal rearrangements, and epigenetic regulation of *TERTp* [2,3,4,10,11,12].

The *TERTp* mutation is commonly found in glioblastomas, IDH-wildtype (GBM-IDHwt) at a rate of 70–90% [13,14,15] and ODG, ranging from 78 to 100% [8,16]. The high incidence of TERTp mutations in both GBM-IDHwt and ODG highlights the significance of this mutation in glioma biology [17]. However, while GBM-IDHwt is associated with a poorer prognosis, ODG typically has a more favorable prognosis, despite having a similar, even higher frequency of *TERTp* mutations [15,18].

Despite data emerging from multiple sources, the prognostic value of *TERTp* mutations in histologically low-grade glioma remains controversial. Under the new guidelines of cIMPACT-NOW update 6 and the 2021 5th edition of WHO classifications of CNS tumors (WHO2021), a diffuse astrocytoma that is IDHwt and lacks high-grade histopathologic features can be diagnosed as GBM-IDHwt if exhibiting any of the following genetic alterations: TERTp mutations, EGFR amplifications, or whole chromosome +7/−10 copy-number changes [19,20,21,22].

However, Berzero et al. focused on low-grade gliomas (histological grade II and III) [23]. This study revealed that patients diagnosed with molecular GBM had a median overall survival (OS) of 17 months compared to those with IDHwt grade II gliomas, not classified as molecular GBM, who had a median OS of 57 months. A significant portion of IDHwt grade II gliomas (16.26, 62%) fulfilled GBM according to the cIMPACT-NOW update 6 criteria, primarily due to isolated *TERTp* mutations; however, the median OS of cases diagnosed with GBM due to these mutations was 88 months, which did not indicate a poorer prognosis. These findings highlight the critical role of both histological grade and molecular profiling in determining the prognosis of IDHwt gliomas and advise caution when classifying IDHwt grade II gliomas as molecular GBMs, particularly when only isolated *TERTp* mutations are present [23]. Richardson et al. emphasized the continued importance of histologic grade in predicting glioma prognosis, presenting a perspective that conflicts with the conclusions of another research group [24]. The conflicting results highlight the complexity of glioma prognosis and underscore the need for further research and consensus in this area, as emphasized by studies from Giannini and Giangaspero, and Olympios et al. [25,26].

IDH mutations, particularly when combined with 1p/19q codeletions, define a subset of gliomas with distinct biological behavior and better prognosis. This is notably evident in ODG, where these genetic alterations are commonly found. The IDH mutations typically suggest a more benign course and responsiveness to therapy, particularly when accompanied by 1p/19q codeletions.

The WHO2021 classification has made significant footsteps in integrating molecular genetic markers into the diagnostic criteria for CNS tumors, including diffuse gliomas. This integration facilitates a more precise classification based on genetic profiles rather than purely histopathological features. This highlights the shift toward a molecularly oriented diagnostic approach, enhancing both the accuracy and prognostic value of glioma classification.

Molecular genetic profiles often offer rapid and actionable insight crucial for diagnosing adult-type diffuse gliomas, enabling precise and immediate classifications and tailoring personalized treatment strategies. However, in the realm of pediatric brain tumors, methylation profiling remains crucial. Pediatric tumors often present with unique and rare subtypes, where methylation patterns provide critical diagnostic and differentiating insights.

This study examined the *TERTp* mutation status within a cohort of patients with adult-type diffuse gliomas as classified by the WHO2021 criteria. The study aimed to evaluate the prognostic significance of both molecular and histologic features of these tumors.

## 2. Materials and Methods

### 2.1. Patient Selection

A total of 528 cases retrieved from the archives of patients with adult-type diffuse gliomas, who underwent surgery at Seoul National University Hospital (SNUH) from 2005 to 2022 were included in this study. All cases initially diagnosed according to WHO2016 or earlier WHO criteria were reclassified based on the updated WHO2021 criteria. Cases that met the criteria for diffuse midline glioma or any pediatric-type glioma were excluded from this study.

This study received approval from the institutional review board of SNUH (IRB No.: C-2203-083-1308) and adheres to the principles outlined in the 1964 Declaration of Helsinki and its subsequent amendments.

### 2.2. Immunohistochemistry

For the immunohistochemistry, 3 μm thick sections were cut from formalin-fixed paraffin-embedded (FFPE) blocks. The following antibodies were utilized: DO-7 monoclonal antibody (mAb) for p53 (1: 1000, DAKO, Glostrup, Denmark), sc48817 for Olig2 (1: 500, Santa Cruz Biotechnology, Santa Cruz, CA, USA), anti-human Ki-67 mAb (1:1000, clone MIB-1, DAKO), ATRX (Merck, St Louis, MO, USA), mAb against the R132H mutation in IDH1 (1:100, clone H09, Dianova, Heidelberg, Germany), and rabbit polyclonal anti-H3 K27M antibody (ABE419, 1:1000, Millipore, Temecula, CA, USA).

The Ki-67 index was counted by morphometric analysis, using the AperioSpectrumPlus n9 algorithm (Leica Biosystems, Wetzlar, Germany) or the Sectra Ki-67 counting algorithm (Sectra, Linköping, Sweden). It was confirmed that there was no significant difference between these two methods.

### 2.3. DNA and RNA Extraction for Molecular Studies

In the microdissection process, representative areas of the tumor, where the tumor cell content exceeded 90%, were delineated on the FFPE sections. DNA and RNA extractions were performed from serial sections using the Maxwell^®^ RSC DNA FFPE Kit (Promega, Madison, WI, USA) and Maxwell^®^ RSC RNA FFPE Kit (AS1440; Promega, Madison, WI, USA) by the manufacturer’s instructions.

The library was generated using SureSelectXT RNA Direct Kit (Agilent, Santa Clara, CA, USA) and sequenced using an Illumina NovaSeq 6000 at Macrogen (Seoul, Republic of Korea).

### 2.4. Fluorescence In Situ Hybridization for 1p and 19q

FFPE blocks underwent fluorescence in situ hybridization (FISH was performed on FFPE blocks. Deletion analysis utilized paired fluorescein isothiocyanate (FITC)/rhodamine-labeled DNA probes specific for chromosome regions 1p (LSI1p36/LSI1q25) and 19q (LSI19q13/LSI19p13). Green and red fluorescent signals were enumerated using a BX01 fluorescence microscope (Olympus, Olympus Corporation, Tokyo, Japan) and the MetaMorph^®^ Imaging System (Universal Imaging, Molecular Devices, San Jose, CA, USA).

For each hybridization, a minimum of 100 non-overlapping nuclei were assessed, and the number of green and red signals was recorded. The deletion was interpreted when the red-to-green ratio was less than 0.8 or more than 50% of nuclei displayed a single red signal. This determination was grounded on the frequency of non-neoplastic nuclei containing one signal (median ± 3 standard deviations) using the same probes in non-neoplastic control (seizure-resection) specimens.

### 2.5. Sanger Sequencing for IDH1/IDH2 Mutation

PCR amplification and Sanger sequencing were conducted using an ABI-PRISM 3730 DNA Analyzer (Applied Biosystems, Vernon Hills, IL, USA). PCR reactions were performed in 40 μL conditions, comprising standard buffer conditions, 200 ng of DNA, and GoTaq DNA Polymerase (Promega, Madison, WI, USA). Sequencing of 2 μL of the PCR product was performed using the BigDye Terminator v3.1 Cycle Sequencing Kit (Applied Biosystems, Foster City, CA, USA).

Twenty-five cycles were performed using 12 ng of the sense primers: *IDH1f* 5′-M13-GTAAAACGACGGCCAGTCGGTCTTCAGAGAAGCCA-3′ or *IDH2f* 5′-GCTGCAGTGGGACCACTATT-3′. The cycling conditions involved denaturation at 96 °C for 10 s, annealing at 50 °C for 5 s, and extension at 60 °C for 4 min.

### 2.6. TERT Promoter Mutation Analysis

The screening for two hotspot mutations, C228T and C250T, in the *TERTp* utilized oligonucleotide primers. PCR amplification of the proximal *TERTp* was conducted using the universal sequencing primer site within M13, with the sequence 5′-GTAAAACGACGGCCAGT-3′, followed by Sanger sequencing using standard methods.

PCR reactions were carried out in 50 μL reaction mixtures comprising 5 μL of DNA, 10 mM of each dNTP, 10 pmole/μL for each primer, 5X Band Doctor™, 10X h-Taq Reaction buffer (15 mM MgCl_2_ mixed), and 2.5 U/μL of Solg™ h-Taq DNA Polymerase. The PCR process began with an initial denaturation at 95 °C for 15 min, followed by 45 cycles of denaturation at 95 °C for 30 s, annealing at 62 °C for 30 s, and extension at 72 °C for 1 min. Finally, a final extension step was performed at 72 °C for 7 min.

### 2.7. Methylation-Specific PCR

Tumor genomic DNA was isolated from FFPE sections for *MGMTp* analysis, followed by bisulfite conversion using the EZ DNA methylation-Gold Kit (Zymo Research, Orange County, CA, USA). Subsequently, methylation-specific PCR for *MGMTp* was conducted using primer pairs designed for methylated and unmethylated *MGMTp* sequences, with the forward and reverse primer sequences being 5′-TTT CGA CGT TCG TAG GTT TTC GC-3′ and 5′-GCA CTC TTC CGA AAA CGA AAC G-3′, respectively.

### 2.8. Next-Generation Sequencing (NGS)

NGS was performed in 398 cases employing the NextSeqTM 550 system (Illumina, San Diego, CA, USA). The analysis utilized a customized brain tumor-targeted gene panel, named FiRST Brain Tumor Panel of SNUH, encompassing 202 to 232 genes and 54 to 155 fusion genes by version up. This comprehensive gene panel, involving both DNA and RNA panels, has received approval from the Korean Ministry of Food and Drug Safety. NGS analysis was conducted as described in our previous article [27]. TERT promoters are included in every gene panel and copy number aberration can be detected. 

### 2.9. Statistical Analysis

The association of clinicopathological parameters was analyzed through Pearson’s chi-square test. The survival rates based on clinical, pathological, and genetic factors were evaluated using Kaplan–Meier survival analysis and log-rank test. Univariate Cox regression analysis was employed to investigate the prognostic significance of clinical and histopathological parameters. The determination of the optimal cutoff for survival analysis was accomplished using the ‘maxstat’ R package (ver. 0.7-25, https://cran.r-project.org/web/packages/maxstat/index.html (accessed on 23 March 2024)). Statistical significance was set at *p*-values < 0.05. All statistical analyses were carried out using SPSS 21 (IBM, Armonk, NY, USA) and R version 3.5.3 (R Foundation, Vienna, Austria).

## 3. Results

### 3.1. Epidemiology and Subgroups of Adult-Type Diffuse Glioma Cohort

A successful reclassification occurred for 528 cases of adult diffuse gliomas following the WHO2021 criteria. Among these, thirteen cases, originally diagnosed as diffuse astrocytoma or anaplastic astrocytoma under the prior WHO diagnostic criteria, were reclassified as GBM-IDHwt. This reclassification was based on the absence of *IDH* mutations and the presence of histopathological features indicative of GBM-IDHwt, such as microvascular proliferation, necrosis, or specific genetic abnormalities, including *TERTp* mutation, *EGFR* amplification, or 7p gain and 10q loss. Among the initial diagnoses of secondary GBM-IDHm, ‘diffuse astrocytoma’, or ‘anaplastic astrocytoma’, 60 tumors with IDH mutations and lacking 1p/19q co-deletion were refined as A-IDHm. Additionally, four cases with *CDKN2A*/*B* homozygous deletion were rediagnosed as A-IDHm, grade G4 (G4), regardless of the absence of evident high-grade histologic features. The diagnosis of ODG remained unchanged in all 68 cases. The details of diagnoses and clinical characteristics of the patients are summarized in Table 1.

### 3.2. TERTp Mutation Status of the Tumors

In GBM-IDHwt, *TERTp*-mutation was present in 240 cases (66.3%). In GBM-IDHwt, the C228T mutation (73.8%) was approximately three times more frequent than the C250T mutation (26.2%).

Among ‘A-IDHm’, six (14.3%) out of the grade 4 tumors and three (7.3%) out of the grade 3 tumors exhibited *TERTp* mutations, whereas no *TERTp* mutations were observed in grade 2 tumors. Specifically, the C228T *TERTp*-mutant was found in four ‘A-IDHm, grade 4’, two in grade 3, while the C250T mutation was found in one in grade 3 and two in grade 4. The occurrence of C228T was twice that of C250T in A-IDHm.

Most patients with ODG displayed *TERTp* mutations, with frequencies of 93.9% in grade 2 and 94.3% in grade 3, respectively. Among ODG cases, the C228T mutation was more prevalent, accounting for 70.3% of cases, approximately twice as common as the C250T mutation, which was observed in 29.7% (Table 2).

### 3.3. Study for Major Molecular Alterations of Adult-Type Diffuse Gliomas

The diagnostic reclassification of these gliomas according to WHO2021 criteria integrated genetic alterations. Among the 98 cases of A-IDHm cases, 3 (3.0%) exhibited *IDH2* mutations, with 2 observed in grade 3 tumors, and 1 in grade 4 tumors.

For *MGMTp* methylation testing, 45.3% (164/362) of GBM-IDHwt, 73.5% (72/98) of A-IDHm, and 95.6% (65/68) of ODG cases showed *MGMTp* methylation positivity. *ATRX* mutation was observed in 4.4% (15/343) of GBM-IDHwt patients, and its frequency in A-IDHm grade 2, grade 3, and grade 4 was 69.2% (9/13), 78.6% (33/42), and 70.3% (26/37), respectively. *TERTp* and *ATRX* mutation status were mutually exclusive, and none of the ODG cases exhibited *ATRX* mutation (Table 3).

For *MYCN* methylation testing, among 369 cases tested, 1 case in 257 GBM-IDHwt, and 3 cases in 3 cases in A-IDHm (2 in grade 4 and 1 in grade 3) showed amplification.

### 3.4. Association of TERTp Mutation with Clinicopathological Features and Major Molecular Alterations

In the present cohort, *TERTp* and *ATRX* mutations were mutually exclusive. Within GBM patients, *TERTp* mutations exhibit no statistically significant association with clinicopathological parameters. Furthermore, *TERTp* mutation demonstrated a statistically significant connection with *EGFR* amplification (*p* < 0.001) but lacked such associations with other mutations like *MGMTp* methylation, *PTEN*, or *CDKN2A*/B deletion. In A-IDHm, and ODG patients, no statistically significant correlation was found between *TERTp* mutations and clinicopathological markers

### 3.5. Overall and Progression-Free Survival Rate

Kaplan–Meier survival curves illustrated significant noteworthy prognostic differences in both OS and progression-free survival (PFS), based on tumor diagnosis and CNS WHO grade (*p* < 0.001) (Figure 1a,b). Specifically within A-IDHm cases, the difference in OS by grade reached statistical significance (*p* < 0.001), although the difference in PFS did not (*p* = 0.296) (Figure 1c,d). On the other hand, among patients with ODG, there was no statistical difference in OS and PFS by histologic grade (*p* = 0.705 and 0.360, respectively) (Figure 1e,f).

### 3.6. Prognostic Impact of TERTp Mutations

The tumor population was divided into two subgroups: Group 1 consisted of GBM-IDHwt and A-IDHm, while Group 2 comprised all ODGs. In Group 1, *TERTp* mutation was a strong prognostic factor (*p* < 0.001) (Figure 2a–d). However, in Group 2, *TERTp* mutation status demonstrated no significant influence on OS (*p*-value cannot be assessed) or PFS (*p* = 0.318) (Figure 3a–d).

No statistically significant difference was observed in OS and PFS between GBM-IDHwt patients with TERTp mutations and those without mutations (Figure 4a,b). However, when GBM-IDHwt patients were classified into four groups according to *TERTp* mutation and *MGMTp* methylation status for a combined analysis, the group with both *TERTp*-mutated and *MGMTp*-unmethylated demonstrated the worst prognosis. This difference reached statistical significance (*p* = 0.018 in OS, *p* = 0.034 in PFS) (Figure 4c,d). On the other hand, when the prognostic impact of *TERTp* was examined separately in *MGMTp*-methylated and *MGMTp*-unmethylated groups, no statistically significant association was found in either group (*p* = 0.259 and 0.231, respectively).

In A-IDHm, grade 4 and grade 3 patients, the presence of *TERTp* mutations did not yield a significant impact on OS (*p* = 0.268 and 0.569) and PFS (*p* = 0.413 and 0.193) (Figure 5). In patients with ‘A-IDHm, grade 2’, the impact of *TERTp* could not be statistically analyzed because no patients harbored *TERTp* mutations. Similarly, in ‘A-IDHm, grade 3’, the small number of patients with *TERTp* mutation (n = 3) precluded a significant analysis.

Additionally, a comparison between hotspot mutations, C228T and C250T, did not reveal significant prognostic differences in any subgroups or disease groups (Figure 2c,d and Figure 3e,f).

### 3.7. Statistical Association between Survival Rate and Clinicopathological Factors in High-Grade Gliomas

We compared the prognosis of patients with GBM-IDHwt with high-grade histologic features such as MVP or necrosis (classic GBM) with that of patients diagnosed with GBM-IDHwt due to the presence of *TERTp* or *EGFR* mutations, although, histologically classified as low-grade (so-called molecular GBM). Patients with molecular GBM tended to have a better prognosis. In other word, there was a tendency for patients with high-grade histology to have a worse prognosis, but this trend did not reach statistical significance (Figure 2e,f).

We evaluated the optimal cutoffs of mitosis and Ki-67 for survival determination in high-grade gliomas. Among GBM-IDHwt patients, those with a mitotic count greater than 11 per 10 high-power fields (HPFs) exhibited a significantly worse prognosis (*p* = 0.029). However, the Ki-67 index did not demonstrate a significant prognostic impact (*p* = 0.234). In contrast, in patients with ‘A-IDHm, G4’, the Ki-67 index had a statistical impact on prognosis when the cutoff was set at 25.33% (*p* = 0.001). The number of mitotic figures did not affect prognosis in A-IDHm, G4 patients (*p* = 0.060).

The impact of several well-established clinicopathological parameters on prognosis was investigated through Kaplan–Meier and Cox regression univariate analyses (Table 4 and Table 5). In the GBM-IDHwt cohort, *EGFR* gene amplification did not exhibit a significant prognostic impact (*p* = 0.919). However, homozygous deletion of *CDKN2A*/*B* (*p* < 0.001) and *PTEN* gene homozygous deletion (*p* = 0.013) were significant poor prognostic factors for both OS and PFS. On the other hand, *MGMTp* methylation (*p* = 0.006) was associated with a favorable prognosis in both OS and PFS. In the multivariate analysis in GBM, *PTEN* alteration was found to be not significant (*p* = 0.219).

For A-IDHm, grade 4, a high Ki-67 index (*p* = 0.003) and *MYCN* amplification (*p* = 0.031) emerged as significantly worse prognostic factors. Unlike GBM, *EGFR* amplification (*p* = 0.607) did not hold significance as a prognostic factor in A-IDHm, grade 4. Similarly, the homozygous deletion of *PTEN* (*p* = 0.501) and *MGMTp* methylation (*p* = 0.174) did not show a significant effect on patient outcomes. In multivariate analysis, *MGMTp* methylation emerged as a better prognostic factor (*p* = 0.024). *CDKN2A*/*2B* was a strong prognostic factor in whole A-IDHm, but within A-IDHm, grade 4, it was not a prognostic factor.

In grade 2 and grade 3 ODGs, where most patients exhibited prolonged OS and PFS rates, the statistical analysis may be less reliable. Consequently, no statistically significant prognostic factors were identified among the clinicopathological findings and molecular markers investigated in these tumor grades.

## 4. Discussion

This study successfully reclassified 528 cases of adult-type diffuse gliomas into distinct gliomas, namely, GBM-IDHwt, A-IDHm, or ODG by the WHO2021 criteria. Analysis of the survival outcomes among these reclassified groups revealed prognostic differences in both OS and PFS based on diagnosis and WHO grade, providing robust support for the validity of the WHO2021 classification.

Among patients previously diagnosed as IDH-mutant GBM, anaplastic astrocytoma, or diffuse astrocytoma, 60 cases exhibited *IDH1* or *IDH2* mutations and were reclassified as A-IDHm. These patients displayed clinicopathological features distinct from IDHwt gliomas despite similar histopathology. In our study, twenty-seven GBM-IDHwt exhibited histopathological features consistent with classic GBM, grade II or III gliomas by WHO2016, with molecular features of GBM, IDHwt, according to the cIMPACT-NOW update 3 and 6 criteria [22,28]. Mortensen et al. indicated that *TERTp* mutation independently plays a prognostic role in IDHwt gliomas lacking high-grade histopathologic features, with survival outcomes comparable to classic GBM-IDHwt [29].

Although molecular GBM tended to exhibit superior OS in our studies than classic GBM, this trend did not reach statistical significance. In our cases, most ‘molecular GBMs’ of our cases only have *TERTp* mutations or EGFR amplifications. However, there were only twenty-seven cases of molecular GBM lacking high-grade histopathological features, and the follow-up duration was 3.7 to 40.8 months (median: 21.9 months), which was insufficient to establish statistical significance. Additional investigations with a larger sample size and extended follow-up periods are necessary to explore this subject further.

We reclassified four cases of A-IDHm, grade 3 to grade 4 based on the presence of the *CDKN2A*/*B* homozygous deletion/mutation, following the WHO2021 criteria [19]. The median OS of grade 2, 3, and 4 A-IDHm patients was 53.0, 44.4, and 35.0 months, respectively. In the A-IDHm patients, relapse or death was experienced 15.4% (2/13) of grade 2 patients, 46.5% (20/43) of grade 3 patients, and 61.9% (26/42) of grade 4 patients. When comparing prognosis based on the reassigned grade within A-IDHm patients, OS exhibited clear stratification by grade, while PFS did not show the same trend. It is worth noting that the small number of cases with recurrence in grade 2 gliomas, and the relatively short follow-up period for the remaining cases (<60 months) may be a confounding factor for accurate statistical analysis, especially in PFS.

Contrarily, WHO2021 does not provide clear criteria for distinguishing between grade 2, and grade 3 ODG. While high cellularity, brisk mitotic activity, microvascular proliferation, and necrosis are indicative of high-grade glioma, the exact criteria remain controversial. Some reports underscore the prognostic relevance of the histologic grade of ODG [30], but the literature has raised questions about the prognostic value [31,32]. In our cohort, there was no statistical difference in OS and PFS based on the histopathological grade of ODG. Recent suggestions propose testing for *CDKN2A*/*B* homozygous deletion to aid in distinguishing grade 2 from grade 3 ODG [33], although this study does not incorporate such considerations.

The *TERTp* mutation frequency in the GBM-IDHwt population examined in this study was 66.3%. While *TERTp* mutations have been reported in the literature to range from as low as 55% to as high as 90%, the frequency identified in this study appears relatively lower compared to other investigations [8,13,17,18,34,35,36,37,38,39,40,41,42,43,44]. It is noteworthy that *TERTp* mutations tend to occur less frequently in East Asian populations compared to North American and European populations. Given that the study is based on a Korean (East Asian) cohort, the frequency aligns with the observed trend [42,44]. In our cohort of A-IDHm, *TERTp* mutations are less common, only 9.2%, while *ATRX* mutations (74%) are more frequently observed compared to GBM-IDHwt, supporting findings from previous reports [17,40,41]. Consistent with earlier studies [8,40,45], *TERTp* mutations were high in our ODGs (94.1%).

The reported frequency of the two hotspot mutations, C228T and C250T is noted to be higher for C228T than C250T in GBM-IDHwt [34,37,46,47]. In our cohort, C228T mutation was observed at an almost threefold higher frequency compared to C250T (73.8%: 26.2%). However, there was no survival difference between the patients with these two hotspot mutations.

The question of whether *TERTp* mutation serves as an independent prognostic factor remains controversial, especially in each diffuse glioma reclassified by the WHO2021 criteria [26,29].

In several previous studies, *TERTp* mutations were not significantly associated with prognosis when adjusting for age and *IDH* mutations in GBM diagnosed according to the 2016 criteria [13,15,37], but in other studies, *TERTp* mutations were associated with the prognosis of GBM patients [18,36,48]. Several investigations have proposed that subgrouping based on a combination of *MGMTp* methylation and *TERTp* mutation status provides significant prognostic value [46,49]. These studies indicate that patients with both *TERTp*-mutation and *MGMTp*-unmethylation tend to experience the worst prognosis [46,50]. The methylation of the *MGMTp* gene, a gene for encoding proteins involved in DNA repair, is associated with a favorable response to Temozolomide treatment in patients with GBM-IDHwt [51]. Although the exact mechanism behind the interaction between *MGMTp* and *TERTp* remains unclear, Arita et al. suggested that the biological consequences of *TERT* activation might influence therapeutic response [46]. Conducting large cohort studies on CNS tumors presents significant challenges due to their rarity, and the presence of numerous confounding factors, such as age, therapeutic intervention, and molecular signatures, further complicates the establishment of the independent prognostic role of *TERTp* mutations [26].

In our cohort, *TERTp* mutations were identified as a significantly worse prognostic factor in Group 1, consisting of GBM-IDHwt and A-IDHm, corroborating the initial findings of our team [14]. However, a thorough analysis using Kaplan–Meier survival curves, Cox regression, and uni- and multivariate analysis indicated that *TERTp* mutations did not significantly affect the prognosis of individual types of diffuse gliomas, GBM-IDHwt and A-IDHm. Gaspar et al.’s meta-analysis, including our institute’s data, showed *TERTp* mutations as poor prognostic factors in A-IDHm across grades 2, 3, and 4 [3]. However, within grade 3 and grade 4 A-IDHm, *TERTp* mutations lacked significance [14]. Notably, grade 4 A-IDHm was previously classified as IDH-mutant GBM under the revised WHO2016 criteria. These findings were further validated in this study with detailed analysis and larger cohorts.

In previous research, a combined analysis of *TERTp* and *MGMTp* methylation status revealed clinically relevant subgroups [46,49]. In our analysis, these four groups exhibited a statistically significant difference in prognosis. However, our findings suggest that the difference between the groups appears to be primarily attributed to the statistical association of *MGMTp* methylation with a favorable prognosis (*p* = 0.006 and *p* = 0.003 in uni- and multivariate analysis). Also, the group with combined *TERTp*-mutations and unmethylated *MGMTp* exhibited the worst prognosis, showing significant effects on OS (*p* = 0.018) and PFS (*p* = 0.034) among GBM-IDHwt patients. Previous research has linked *TERTp* mutations with poor prognosis in *IDH*-mutant high-grade gliomas [17,52]. However, some studies have suggested a potential favorable impact of these mutations in grade 2 A-IDHm patients, indicating opposing prognostic effects depending on the tumor grade within the same tumor type [18,41,52]. Research on this subgroup is limited due to the small sizes of cohorts involved, necessitating further investigations in larger cohorts.

Assessing the statistical impact in ODGs (Group 2) was challenging due to the high frequency of *TERTp* mutations (94.1%) and the prolonged survival of patients. Some studies proposed *TERTp* as a favorable prognostic factor in gliomas with *IDH* mutations, regardless of 1p/19q co-deletion [45,52].

The separate analysis of C228T and C250T hotspot mutations showed no significant effect on OS and PFS, suggesting similar roles in cancer development and prognosis due to their identical E-twenty six1 (ETS1) binding motifs structures [53].

In both univariate and multivariate analyses in GBM, age, *MGMTp* methylation, and *CDKN2A*/B homozygous deletion were statistically significant prognostic factors, while in univariate analysis in A-IDHm, grade 4, the Ki-67 index and *MYCN* amplifications were statistically significant prognostic factors. In multivariate analysis, *MGMTp* methylation was included among these poor prognostic factors.

Considering the high prevalence of *TERTp* mutation, its presence alone is sufficient for a grade 4 diagnosis, making targeting telomerase a promising anticancer strategy [54]. Several preclinical studies have demonstrated that inhibiting telomerase activity can significantly reduce tumor cell proliferation, including in glioma [55,56,57]. Clinical trials of telomerase inhibitors have been attempted across various solid tumors and hematologic malignancies [58,59,60,61].

Further preclinical research has explored innovative methods to silence the *TERTp*. These methods include programmable base editing techniques that directly modify the *TERTp* mutation [57,62], CRISPR-mediated targeting of GA-binding transcription factors that regulate TERT expression [63], and the use of 6-thio-2′-deoxyguanosine [64,65], a telomerase substrate precursor analog, which has shown efficacy in inhibiting tumor cell growth in glioma models.

However, the transition of these therapies from bench to bedside in HGG treatment presents significant challenges. Any therapeutic agent must be able to cross the blood–brain barrier and minimize systemic side effects to be considered viable for clinical use in these patients. Therefore, while the potential of these strategies is significant, their practical application requires considerable further research.

This study is subject to several limitations. Firstly, its retrospective design and conduct at a single institution may limit the generalizability of the findings. To confirm these results, a multicenter prospective study is necessary. Additionally, while we employed an NGS panel that covers the two hotspot mutations in the *TERTp*, we did not investigate other known telomere maintenance mechanisms that can activate telomerase, such as *TERT* amplification or *TERC* (Telomerase RNA Component) gene amplification, chromosomal rearrangement or epigenetic modifications of *TERT* gene, altered length of telomere, and non-defined telomere maintenance mechanism [3]. Recent research has highlighted the role of epigenetic factors, including *TERTp* methylation [10,11,66]. Addressing these additional mechanisms in future studies will be crucial to develop a more comprehensive understanding of telomerase activation in cancer.

## 5. Conclusions

This study confirmed that the classification of diffuse gliomas according to the WHO2021 criteria effectively reflected prognosis, demonstrating significant differences in molecular profiles and their association with prognosis based on IDH mutation status or 1p/19q co-deletion. Specifically, *TERTp* mutation was identified as a predictor of poor prognosis in astrocytic tumors (group 1), which includes A-IDHm, and GBM-IDHwt. However, within the classifications of GBM-IDHwt, A-IDHm, and ODG based on genetics-integrated diagnosis according to the WHO2021 criteria, *TERTp* mutations did not stand out as a prognostic factor. 

Additionally, *MGMTp* methylation was associated with a favorable prognosis in both univariate and multivariate analyses. Notably, the subgroup harboring both *TERTp*-mutation and unmethylated *MGMTp* exhibited the worst prognosis in OS and PFS among GBM-IDHwt patients. These findings highlight the importance of considering combined pathological and molecular factors for more precise tumor classification and better prognostic assessment in patients.

## Figures and Tables

**Figure 1 cancers-16-02032-f001:**
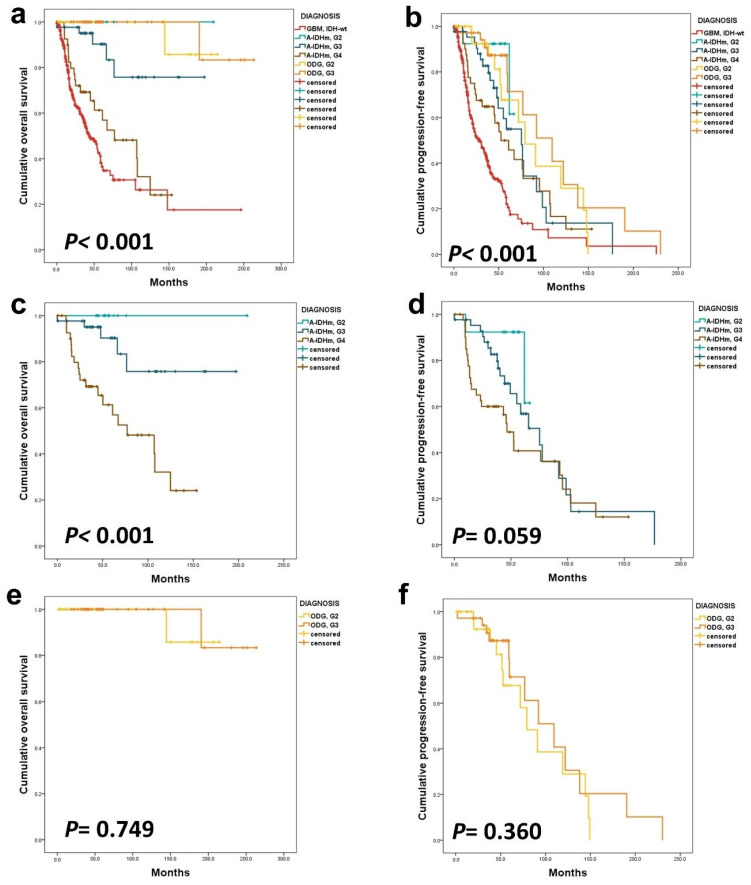
The Kaplan–Meier survival analysis was conducted on adult-type diffuse gliomas. In this cohort, all gliomas exhibited significant differences in OS (**a**) and PFS (**b**). Within the A-IDHm group, there were statistically different OS (**c**) based on CNS WHO grade, but PFS (**d**) did not show a significant difference. In the ODG group, histologic grade only impacted patients’ PFS (**e**,**f**).

**Figure 2 cancers-16-02032-f002:**
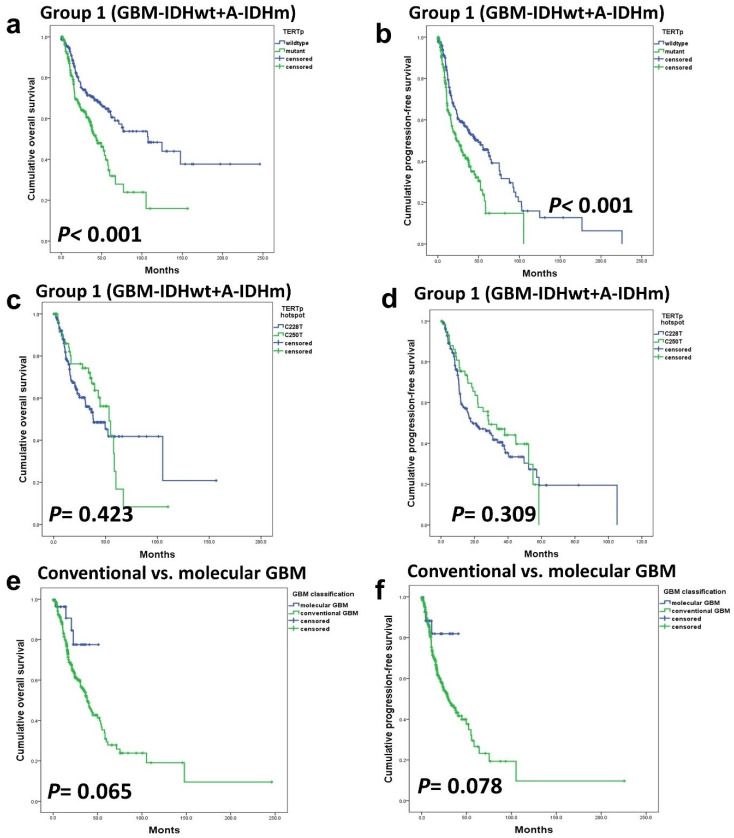
Within group 1, patients with astrocytic tumors, encompassing A-IDHm and GBM-IDHwt *TERTp* mutations had a significant impact on OS (**a**) and PFS (**b**). There was no statistically significant difference in survival between *TERTp* hotspot C228T and C250T mutations (**c**,**d**). There were no significant prognostic differences between classic and molecular GBM (**e**,**f**).

**Figure 3 cancers-16-02032-f003:**
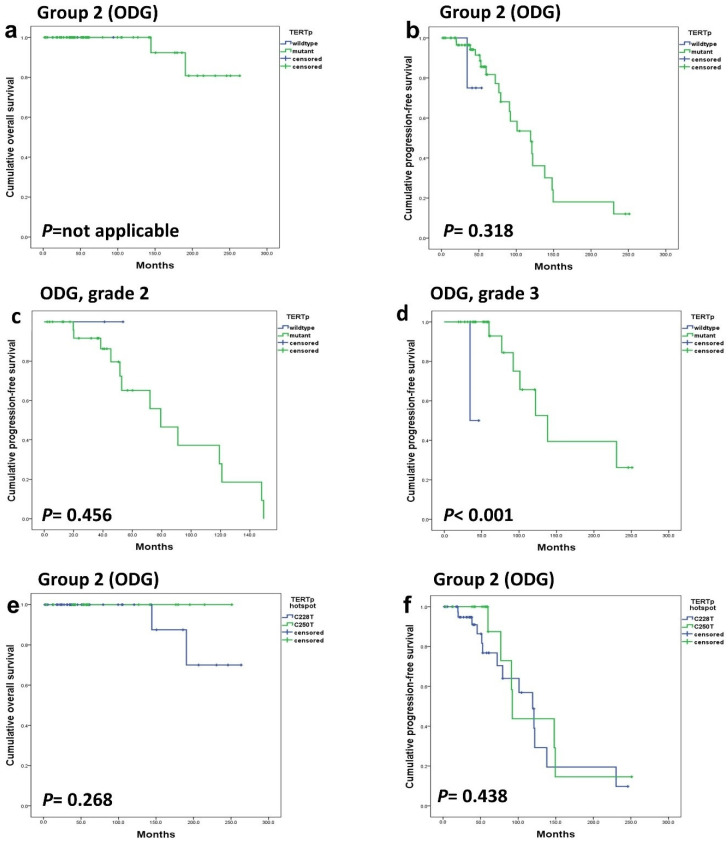
In group 2 (ODGs), both OS (**a**) and PFS (**b**) were not affected by the *TERTp* mutation. *TERTp* mutation did not affect the PFS of grade 2 ODG (**c**) but had a significant effect on PFS in grade 3 ODG (**d**). Additionally, the hotspot mutation type did not exhibit a significant association with OS (**e**) and PFS (**f**).

**Figure 4 cancers-16-02032-f004:**
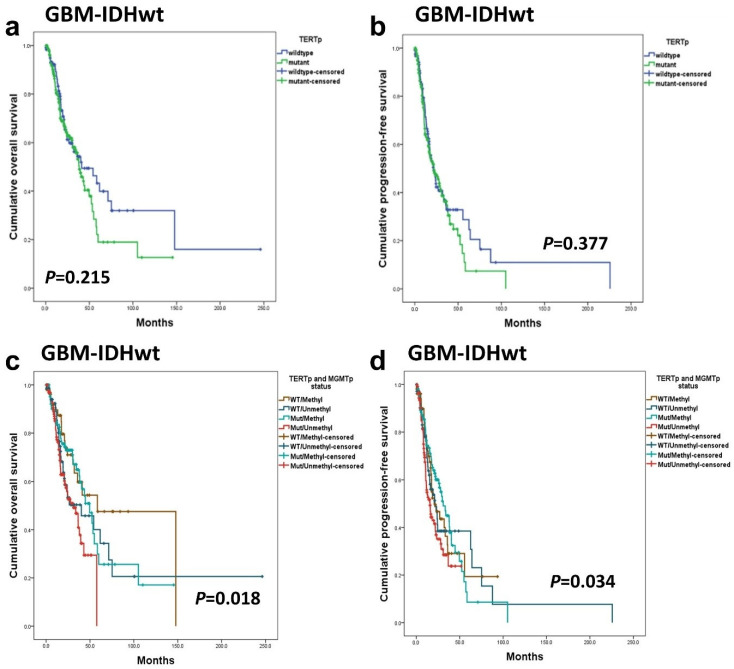
In GBM-IDHwt patients, *TERTp* mutation was not associated with OS (**a**) and PFS (**b**). When GBM-IDHwt patients were divided into four groups on *TERTp* mutation and *MGMTp* methylation status, the Kaplan–Meier survival analysis revealed statistically significant differences in OS (**c**) and PFS (**d**).

**Figure 5 cancers-16-02032-f005:**
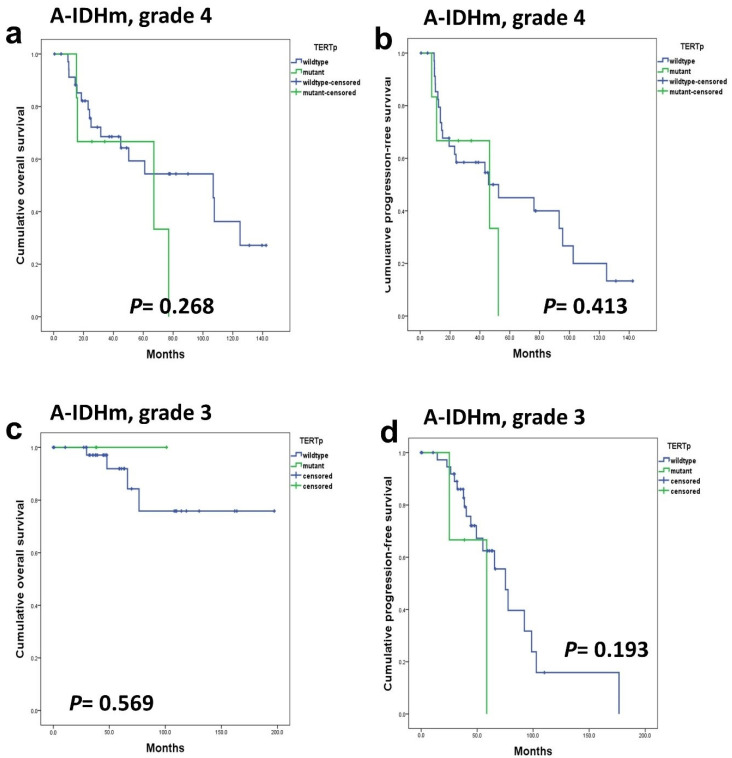
*TERTp* mutations did not have a prognostic impact on OS (**a**,**c**) and PFS (**b**,**d**) in A-IDHm, grade 4 patients and grade 3 patients.

**Table 1 cancers-16-02032-t001:** The Data distribution of the patient group with diffuse gliomas comprised 362 cases of GBM, IDH-wildtype, 98 astrocytomas, IDH-mutant, and 68 cases of oligodendrogliomas.

Items	Variables	Classical GBM-IDHwt n = 335	Molecular GBM-IDHwt n = 27	Total GBM-IDHwt n = 362	A-IDHm, G2 n = 13	A-IDHm, G3 n = 43	A-IDHm, G4 n = 42	Total A-IDHm n = 98	ODG, G2 n = 33	ODG, G3 n = 35	Total ODG n = 68
Age	Range (median)	2–86 (60)	36–78 (56)	2–86 (60)	26–55 (47)	18–66 (36)	27–70 (39.5)	18–70 (38)	22–66 (42)	31–69 (52)	22–69 (49)
Gender	Male	181	12	193	9	22	27	58	20	20	40
	Female	154	15	169	4	21	15	40	13	15	28
OS (months)	Range (median)	0.3–236.2 (38.3)	2.2–50.9 (20.9)	0.2–246.2 (40.2)	42.8–209.4 (NA/56.0)	0.8–197.2 (NA/47.5)	0.6–153.6 (77.0/37.1)	0.6–209.4 (NA/47.1)	2.0–214.7 (NA/42.7)	19.1–263.4 (NA/56.5)	2.0–263.4 (NA/52.5)
PFS (months)	Range (median)	0.2–225.7 (30.0)	2.2–40.8 (11.2)	0.2–225.7 (22.0)	10.0–61.8 (NA/53.0)	0.8–176.7 (75.3/44.4)	0.6–153.6 (46.5/35.0)	0.6–176.6 (65.5/43.5)	2.0–149.4 (79.3)	19.1–251.0 (109.5)	2.0–251.0 (92.2)
Mitosis	Range (mean)	1–210 (32.4)	0–72 (15.3)	0–250 (31.1)	0–2 (0.9)	3–60 (11.0)	0–110 (25.0)	0–110 (15.7)	0–12 (3.5)	1–55 (15.1)	0–55 (9.5)
Ki-67 index (%)	Range (mean)	1.0–96.9 (40.8)	1.0–92.4 (20.0)	1.0–96. 9 (39.2)	0.2–4.3 (2.6)	1.0–46.0 (9.7)	1.7–90.2 (31.6)	0.2–90.2 (18.2)	0.9–26.6 (7.0)	7.6–74.3 (29.6)	0.9–74.3 (18.6)

Abbreviations: GBM-IDHwt, GBM, IDH-wildtype; A-IDHm, Astrocytoma-IDHm; G2, CNS WHO grade 2; G3, CNS WHO grade 3; G4, CNS WHO grade 4; ODG, Oligodendroglioma; OS, overall survival; PFS, progression-free survival; NA, not available by SPSS/median by excel.

**Table 2 cancers-16-02032-t002:** The frequency of *TERT* promoter mutation status of each glioma.

*TERTp*	GBM-IDHwt n = 362	A-IDHm, G2 n = 13	A-IDHm, G3 n = 43	A-IDHm, G4 n = 42	A-IDHm n = 98	ODG, G2 n = 33	ODG, G3 n = 35	ODG n = 68
Mutant (%)	240/362 (66.3)	0/13 (0)	3/43 (7.0)	6/42 (14.3)	9/98 (9.2)	31/33 (93.9)	33/35 (94.3)	64/68 (94.1)
C228T	177/240 (73.8)	0/0 (0)	2/3 (66.7)	4/6 (66.7)	6/9 (66.6)	23/31 (74.2)	22/33 (66.7)	45/64 (70.3)
C250T	63/240 (26.2)	0/0 (0)	1/3 (33.3)	2/6 (33.3)	3/9 (33.3)	8/31 (25.8)	11/33 (33.3)	19/64 (29.7)

Abbreviations: *TERTp*, *TERT* promoter.

**Table 3 cancers-16-02032-t003:** Major genetic alterations observed in adult-type diffuse gliomas of SNUH cases.

Genes	Changes	Conventional GBM	Molecular GBM	GBM-IDHwt n = 364 (%)	A-IDHm, G2 n = 13 (%)	A-IDHm, G3 n = 43 (%)	A-IDHm, G4 n = 42 (%)	Total A-IDHm n = 98 (%)	ODG, G2 n = 33 (%)	ODG, G3 n = 35 (%)	Total ODG n = 68 (%)
MGMTp	Methylated	159/335 (47.5)	5/27 (18.5)	164/364 (45.3)	8/13 (61.5)	30/43 (69.8)	34/42 (81.0)	72/98 (73.5)	32/33 (97.0)	33/35 (94.3)	65/68 (95.6)
	Unmethylated	176/335 (52.5)	22/27 (81.5)	198/362 (54.7)	5/13 (38.5)	13/43 (30.2)	8/42 (19.0)	26/98 (26.5)	1/33 (3.0)	2/35 (5.7)	3/68 (4.4)
ATRX	Mutant	14/316 (4.4)	1/27 (3.7)	15/343 (4.4)	9/13 (69.2)	33/42 (78.6)	26/37 (70.3)	68/92 (73.9)	0/33 (0)	0/35 (0)	0/68 (0)
	Wildtype	302/316 (95.6)	26/27 (96.3)	328/343 (95.6)	4/13 (30.8)	9/42 (21.4)	11/37 (29.7)	24/92 (26.1)	33/33 (100)	35/35 (100)	68/68 (100)
1p/19q	Co-deletion	0/330 (0)	0/26 (0)	0/356 (0)	0/13 (0)	0/43 (0)	0/41 (0)	0/98 (0)	33/33 (100)	35/35 (100)	68/68 (100)
	No co-deletion	330/330 (100)	26/26 (100)	356/356 (100)	13/13 (100)	43/43 (100)	42/42 (100)	98/98 (100)	0/33 (0)	0/35 (0)	0/68 (0)
EGFR	Amplification	90/334 (26.9)	9/27 (33.3)	99/361 (27.4)	0/13 (0)	0/43 (0)	2/42 (4.8)	2/98 (2.0)	0/33 (0)	0/35 (0)	0/68 (0)
	No amplification	244/334 (73.1)	18/27(66.7)	262/361 (72.6)	13/13 (100)	43/43 (0)	40/42 (95.2)	96/98 (98.0)	33/33 (100)	35/35 (100)	68/68 (100)
CDKN2A/B	Deletion	165/319 (51.7)	9/27 (33.3)	174/346 (10.3)	0/13 (0)	0/41 (0)	20/42 (47.6)	20/96 (20.8)	1/33 (3.0)	7/35 (20.0)	5/68 (7.4)
	No deletion	154/319 (48.3)	18/27 (66.7)	172/346 (49.7)	13/13 (100)	41/41 (100)	22/42 (52.4)	76/96 (79.2)	32/33 (97.0)	28/35 (80.0)	63/68 (92.6)
PTEN	Deletion	57/334 (17.1)	2/27 (7.4)	59/361 (16.3)	0/13 (0)	2/43 (4.7)	6/42 (14.3)	8/98 (8.2)	0/33 (0)	0/35 (11.4)	0/68 (0)
	No deletion	277/334 (82.9)	25/27 (92.6)	302/361 (83.7)	13/13 (100)	41/43 (95.3)	36/42 (85.7)	90/98 (91.8)	33/33 (100)	35/35 (88.6)	68/68 (100)

**Table 4 cancers-16-02032-t004:** Univariate and multivariate analysis of clinicopathological parameters affecting overall survivals in patients with GBM-IDHwt.

Variables	Comparison	Univariate		
		*p*	HR	95%CI
*TERTp*	Wildtype vs. Mutant	0.211	0.799	0.561–1.140
Age	Age ≥ 55 vs. <55	0.018	1.530	1.074–2.178
Gender	Male vs. Female	0.770	0.952	0.685–1.323
MVP/necrosis	Absent vs. Present	0.074	0.636	0.386–1.046
Mitosis	Mitosis ≥ 11 vs. <11	0.021	1.607	1.073–2.406
Ki-67 (cutoff > 11.19%)	High vs. Low	0.195	1.463	0.195–1.463
*MGMTp*	Methylated vs. Unmethylated	0.006	1.166	1.069–1.498
*ATRX*	Wildtype vs. Mutant	0.546	1.125	0.768–1.648
*EGFR* amplification	Absent vs. Present	0.919	0.991	0.826–1.188
*CDKN2A*/B homozygous deletion	Absent vs. Present	<0.001	0.679	0.566–0.816
*PTEN* loss	Absent vs. Present	0.013	0.781	0.642–0.949
**Variables**	**Comparison**	**Multivariate**		
		** *p* **	**HR**	**95%CI**
*TERTp*	Wildtype vs. Mutant	0.156	1.394	0.881–2.207
Age	Age ≥ 55 vs. <55	0.002	1.023	1.009–1.038
Gender	Male vs. Female	0.733	0.935	0.634–1.377
MVP/necrosis	Absent vs. Present	0.147	0.440	0.145–1.334
Mitosis	Mitosis ≥ 11 vs. <11	0.170	1.463	0.849–2.519
Ki-67 (cutoff > 11.19%)	High vs. Low	0.585	0.798	0.354–1.797
*MGMTp*	Methylated vs. Unmethylated	0.003	1.851	1.237–2.769
*ATRX*	Wildtype vs. Mutant	0.378	1.466	0.626–3.436
*EGFR* amplification	Absent vs. Present	0.129	1.434	0.900–2.284
*CDKN2A*/B homozygous deletion	Absent vs. Present	0.001	0.501	0.328–0.765
*PTEN* loss	Absent vs. Present	0.219	0.754	0.480–1.183

**Table 5 cancers-16-02032-t005:** Univariate and multivariate analysis of clinicopathological parameters affecting overall survival in patients with astrocytoma, IDH-mutant, CNS WHO grade 4.

Variables	Comparison	Univariate		
		*p*	HR	95%CI
*TERTp* mutation	Wildtype vs. Mutant	0.275	1.869	0.308–5.750
Age	Age ≥ 55 vs. <55	0.896	0.929	0.309–2.796
Gender	Male vs. Female	0.376	0.663	0.267–1.646
MVP/necrosis	Absent vs. Present	0.649	1.348	0.373–4.862
Mitosis	Mitosis ≥ 32 vs. <32	0.130	2.001	0.815–4.913
Ki-67 (cutoff > 25.3%)	High vs. Low	0.003	4.578	1.653–12.673
*MGMTp*	Methylated vs. Unmethylated	0.174	2.045	0.730–5.726
*ATRX*	Wildtype vs. Mutant	0.217	0.544	0.207–1.429
*EGFR* amplification	Absent vs. Present	0.607	1.704	0.226–12.981
*CDKN2A*/B homozygous deletion	Absent vs. Present	0.345	0.648	0.264–1.594
*PTEN* deletion	Absent vs. Present	0.501	1.479	0.473–4.627
*MYCN* amplification	Absent vs. Present	0.031	0.137	0.023–0.832
**Variables**	**Comparison**	**Multivariate**		
		** *p* **	**HR**	**95%CI**
*TERTp* mutation	Wildtype vs. Mutant	0.106	4.376	0.729–26.267
Age	Age ≥ 55 vs. <55	0.073	3.594	0.886–14.580
Gender	Male vs. Female	0.359	0.488	0.105–2.262
MVP/necrosis	Absent vs. Present	0.538	1.635	0.342–7.813
Mitosis	Mitosis ≥ 32 vs. <32	0.038	0.168	0.031–0.906
Ki-67 (cutoff > 25.3%)	High vs. Low	0.002	16.567	2.872–95.576
*MGMTp*	Methylated vs. Unmethylated	0.024	6.587	1.275–34.038
*ATRX*	Wildtype vs. Mutant	0.604	0.694	0.174–2.767
*EGFR* amplification	Absent vs. Present	0.976	0.965	0.094–10.227
*CDKN2A*/B homozygous deletion	Absent vs. Present	0.950	0.954	0.220–4.130
*PTEN* deletion	Absent vs. Present	0.746	1.376	0.199–9.538
*MYCN* amplification	Absent vs. Present	-	-	-

## Data Availability

The datasets used and/or analyzed during the current study are available from the corresponding author upon reasonable request.

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
