# Peer review of "Prognostic Impact of TERT Promoter Mutations in Adult-Type Diffuse Gliomas Based on WHO2021 Criteria"

_cancers, 2024, doi:10.3390/cancers16112032_

Round 1

Reviewer 1 Report

Comments and Suggestions for Authors

In the present study, Lee et al. reclassified 528 cases of adult-type diffuse glioma from a single center according to the criteria of the latest classification of the WHO2021. They also analyzed the presence of TERT promoter mutation in the different subgroups and its correlation with other markers and the impact on the OS and PFS. The study is important, and the manuscript is well written. However, I have some comments that need to be addressed.

Major comments:

-        In the Introduction section, a section about TERT gene and its promoter should be added. This is very important for the non-expert and non-telomere people so they can understand the role, function and physiopathology.

There are some useful references to help you :

To introduce TERT gene and telomerase, and telomeres maintenance (PMID: 29751586, 11102810) and the complex regulation (PMID 36833366)

To mention the mechanism of TERT reactivation in cancer, including TERTp mutation (PMID 23348506, 23887712), amplification, locus rearrangement (30760854) and more recently through epigenetic regulation via promoter methylation (PMID 33715271, 30358567)

To say that these mechanisms are reported in several cancers (PMID 29526163, 34065134)

-        TABLE 2: Please recheck the calculations of TERT promoter mutation % : i.e. in the total ODG, 64 out of 68 presented a TERTp mutation, and this is a 94.1 % and not 98.5% as mentioned in the table. Please recheck and correct all the numbers and calculations in the table and text. And if applicable, recheck the statistical tests and values.

-        Among different histologic grades of ODG patients, the P-value regarding the PFS in the text is different than the graphs: 0.012 line 248 and 0.360 in the fig 1.e. Please recheck the values and correct the wrong one.

-        In the discussion and conclusion, there is no interpretation of TERTp mutation in light of the clinical and therapeutic implications. What could be the role or effect of the presence of TERTp mutation on the outcome of the treatments?

-        In case of absence of TERTp mutations, did the authors check the literature or investigate if an epigenetic mechanism could be present and thus leading also to TERT activation?

Minor comments:

-        The last 2 paragraphs of the introduction need to be restructured in a way that the previous work is mentioned first, then the current investigation.

-        For the grades of the tumor, as recommended in the newest version of classification please use Arabic numerals only (i.e. line 45 and 78 Arabic numerals, while line 68 Roman numerals)

-        Which kit was used for RNA extraction? Only the DNA extraction kit was mentioned.

-        Suggestion, Line 171-172 : “TERT promoter mutations are not covered by NGS panels”

-        Table 3 seems to be duplicated.

-        Line 232 remove the word mutation.

-        Fig 4a and 4b : P-values are written twice on the graphs. Please keep only one value.

-        Table 4 and 5 should be moved to the correct section of the results and not in the discussion.  

-        Line 372: “it is” instead of “it’s”. The authors should avoid using contractions.

-        Line 392-393: “The MGMTp gene”, better to say “The promoter of the MGMT gene”

Comments on the Quality of English Language

minor editing

Author Response

Thank you for your kind review and comments.

Reviewer 1

In the present study, Lee et al. reclassified 528 cases of adult-type diffuse glioma from a single center according to the criteria of the latest classification of the WHO2021. They also analyzed the presence of TERT promoter mutation in the different subgroups and its correlation with other markers and the impact on the OS and PFS. The study is important, and the manuscript is well written. However, I have some comments that need to be addressed.

Major comments:

-        In the Introduction section, a section about TERT gene and its promoter should be added. This is very important for the non-expert and non-telomere people so they can understand the role, function and physiopathology.

There are some useful references to help you:

To introduce TERT gene and telomerase, and telomeres maintenance (PMID: 29751586, 11102810) and the complex regulation (PMID 36833366)

To mention the mechanism of TERT reactivation in cancer, including TERTp mutation (PMID 23348506, 23887712), amplification, locus rearrangement (30760854) and more recently through epigenetic regulation via promoter methylation (PMID 33715271, 30358567)

To say that these mechanisms are reported in several cancers (PMID 29526163, 34065134)

RESPONSE

We have added a section on telomerase and related references to the first paragraph of the Introduction section as suggested.

-        TABLE 2: Please recheck the calculations of TERT promoter mutation % : i.e. in the total ODG, 64 out of 68 presented a TERTp mutation, and this is a 94.1 % and not 98.5% as mentioned in the table. Please recheck and correct all the numbers and calculations in the table and text. And if applicable, recheck the statistical tests and values.

RESPONSE

We rechecked the calculations and corrected the number error (98.5% to 94.1%).

-        Among different histologic grades of ODG patients, the P-value regarding the PFS in the text is different than the graphs: 0.012 line 248 and 0.360 in the fig 1.e. Please recheck the values and correct the wrong one.

RESPONSE

We corrected the incorrect P-value of 0.012 in the manuscript to the correct value of 0.360 and revised the description in the manuscript.

-        In the discussion and conclusion, there is no interpretation of TERTp mutation in light of the clinical and therapeutic implications. What could be the role or effect of the presence of TERTp mutation on the outcome of the treatments?

RESPONSE

It is well known that there is interest in therapeutics targeting telomerase or related mediators in managing gliomas. Several anticancer agents have been proposed, many of which are being studied in the preclinical stage. In our study, patients with high-grade glioma were treated with temozolomide after surgical resection, and the correlation between TERTp mutation status and treatment response was unclear. We have added interpretation from the clinical perspective to the Discussion section.

-        In case of absence of TERTp mutations, did the authors check the literature or investigate if an epigenetic mechanism could be present and thus leading also to TERT activation?

RESPONSE

The fact that tumor cells that do not harbor TERTp hotspot mutations can maintain telomerase activity by alternative TERT upregulation mechanisms may explain the lack of prognostic value of hotspot mutation status in this study. We only examined TERTp hotspot mutations, and other known mechanisms that drive telomerase activity, such as TERTp amplification, chromosomal rearrangement, or epigenetic changes, were not investigated. We have added the limitations of this study mentioned above to the Discussion section and anticipate that follow-up studies will reflect additional known representative TERTp mutations and epigenetic changes.

Minor comments:

-        The last 2 paragraphs of the introduction need to be restructured in a way that the previous work is mentioned first, then the current investigation.

RESPONSE

We have repositioned the two paragraphs.

-        For the grades of the tumor, as recommended in the newest version of classification please use Arabic numerals only (i.e. line 45 and 78 Arabic numerals, while line 68 Roman numerals)

RESPONSE

The Roman numerals in line 68 were used while citing a paper by Berzero et al. The paper was published in 2021, and their study used the 2016 WHO classification. To emphasize that point, we did not replace the Roman numeral with Arabic numerals.

-        Which kit was used for RNA extraction? Only the DNA extraction kit was mentioned.

RESPONSE

We appreciate for pointing out the missing statement. We have added the mention of RNA extraction kit to the Methods section.

-        Suggestion, Line 171-172 : “TERT promoter mutations are not covered by NGS panels”

RESPONSE

The NGS panel utilized in this study specifically includes coverage for mutations in the TERT promoter. Notably, the mutations such as C228T and C250T in the TERT promoter are covered by our FiRTST panel. Detailed information on these mutations and others included in the panel can be found in Supplementary Table 1.

-        Table 3 seems to be duplicated.

RESPONSE

We deleted one of the duplicate tables.

-        Line 232 remove the word mutation.

RESPONSE

We have removed the word.

-        Fig 4a and 4b : P-values are written twice on the graphs. Please keep only one value.

RESPONSE

Thank you for pointing out the redundancy in Figures 4a and 4b. We ensure that each graph only displays a single p-value to maintain clarity and prevent any confusion.

-        Table 4 and 5 should be moved to the correct section of the results and not in the discussion. 

RESPONSE

Thank you for your feedback regarding the placement of Table 4 and Table 5. I agree that for better clarity and coherence, we moved these tables in the appropriate section of the results rather than in the discussion.

.

-        Line 372: “it is” instead of “it’s”. The authors should avoid using contractions.

RESPONSE

We changed the contraction to a formal form as suggested.

-        Line 392-393: “The MGMTp gene”, better to say “The promoter of the MGMT gene”

RESPONSE

We have revised the description as suggested.

Reviewer 2 Report

Comments and Suggestions for Authors

This study presented the patient survival of brain tumor subtypes together TERT promoter mutation status in their own retrospective collection of over 500 glioma patients. Although the question is interesting, the quality of the work was poor.  For example, the classification of glioma needs to be verified by standard practice using DNA methylation arrays. The lack of prognosis power of TERT promoter mutations is problematic and worth scrutinizing. The methylation status of distal promoter (known as THOR) needs to be evaluated as it has been showed by multiple studies as prognostic in multiple tumors.  The main conclusion of glioma subtypes showing different prognosis is not novel. Therefore, the reviewer suggests more rigorous and in-depth analyses to identify potential interaction glioma subtypes and TERT promoter mutation/methylation with patient survival.

Author Response

Thank you for your kind review and comments

Reviewer 2

This study presented the patient survival of brain tumor subtypes together TERT promoter mutation status in their own retrospective collection of over 500 glioma patients. Although the question is interesting, the quality of the work was poor.  For example, the classification of glioma needs to be verified by standard practice using DNA methylation arrays. The lack of prognosis power of TERT promoter mutations is problematic and worth scrutinizing. The methylation status of distal promoter (known as THOR) needs to be evaluated as it has been showed by multiple studies as prognostic in multiple tumors.  The main conclusion of glioma subtypes showing different prognosis is not novel. Therefore, the reviewer suggests more rigorous and in-depth analyses to identify potential interaction glioma subtypes and TERT promoter mutation/methylation with patient survival.

RESPONSE

As mentioned in the manuscript, the prognostic significance of TERTp hotspot mutations is still a topic of debate. However, we do not consider our study to be unreliable just because the TERTp mutation did not show statistical significance in predicting the prognosis of glioblastoma patients.

In our study, we intentionally excluded cases that were ambiguous and might have required a methylation study for definitive diagnosis. According to the WHO2021 guidelines, while DNA methylation arrays are recommended, they are not mandatory for the classification and diagnosis of adult-type diffuse gliomas. In practical clinical settings, it is neither necessary nor feasible to validate every diagnosis when it is already clear.

We acknowledge that our study did not explore mutations in the TERT promoter beyond the two known hotspot mutations, nor did it examine epigenetic changes in TERTp known to enhance TERT expression. We recognize this as a limitation of our current research and an area for future investigation. We have noted this point in the Discussion section to acknowledge the reviewer's valuable feedback.
